# Unconstrained Monitoring Method for Heartbeat Signals Measurement using Pressure Sensors Array

**DOI:** 10.3390/s19020368

**Published:** 2019-01-17

**Authors:** Yongxiang Jiang, Sanpeng Deng, Hongchang Sun, Yuming Qi

**Affiliations:** Tianjin University of Technology and Education, Institute of Robotics and Intelligent Equipment, Tianjin 300222, China; sanpeng@yeah.net (S.D.); sunhongchang@tute.edu.cn (H.S.); chigym@163.com (Y.Q.)

**Keywords:** unconstrained heartbeat signal extraction, pressure sensors array, extreme learning machine (ELM), correlation coefficient

## Abstract

An unconstrained monitoring method for a driver’s heartbeat is investigated in this paper. Signal measurement was carried out by using pressure sensors array. Due to the inevitable changes of posture during driving, the monitoring place for heartbeat measurement needs to be adjusted accordingly. An experiment was conducted to attach a pressure sensors array to the backrest of a seat. On the basis of the extreme learning machine classification method, driving posture can be recognized by monitoring the distribution of pressure signals. Then, a band-pass filter in heart rate range is adapted to the pressure signals in the frequency domain. Furthermore, a peak point array of the processed pressure frequency spectrum is derived and has the same distribution as the pressure signals. Thus, the heartbeat signals can be extracted from pressure sensors. Then, the correlation coefficient analysis of heartbeat signals and electrocardio-signals is performed. The results show a high level of correlation. Finally, the effects of driving posture on heartbeat signal extraction are discussed to obtain a theoretical foundation for measuring point real-time adjustment.

## 1. Introduction

Over 1.2 million people die in traffic accidents every year and driver fatigue is a significant factor causing traffic accidents [1]. Thus, driver fatigue monitoring has been a popular research topic since the 1990s. Driving fatigue includes three categories: (1) Periodic fatigue caused by insufficient sleep and overtime work; (2) physiological fatigue caused by static force work of muscle; and (3) psychological fatigue caused by high intense of nerve centre [2]. Accordingly, currently developed monitoring methods can be divided as follows: (1) Visual-based fatigue monitoring, such as blinking, nodding, facial expression changes, head movement, eye motion; (2) driving actions, such as behaviors in steering, braking, acceleration, and body “slumping” or body shifting; and (3) physiological information, including electroencephalo-, electromyo-, and electrocardio-signals, and respiration [3].

Physiological information for physiological fatigue monitoring was confirmed to have a relatively high accuracy [4]. An electroencephalo-signal is considered a gold standard for measuring mental and physical fatigue [5]. Significant changes in Electromyo [6] and heartbeat [7] can also be observed under fatigue conditions. However, the traditional monitoring method for physiological information must set electrodes to the driver, thereby constraining the driving movement. The unconstrained heartbeat signal monitoring is the most feasible method that has been investigated in recent years with encouraging results.

The first type of research in unconstrained heartbeat signal monitoring is based on the Doppler effect. The heart undergoes volumetric changes during each cardiac cycle while pumping blood through the cardiovascular system. These changes are then reflected in the periodical movement of the chest. Frequencies between 0.5 and 2 Hz are due to the heartbeat. Radar devices enable the detection of motion with considerably small scales. Thus, heartbeat signals can be extracted [8]. However, several studies have demonstrated that accuracy is significantly affected by ambient body movements and an inevitable vibration in the driving process [9]. Inspired by the radar indoor localization technology, Wi-Fi radio frequency signals have been used as a non-intrusive environmental sensing tool. RF signals propagate in the wireless medium through a multipath, bouncing off different objects before arriving at a receiver and, hence, carrying information about the environment. Centimeter-scale human activity, such as respiration rate, can be measured [10]. The research developed another promising path for unconstrained heartbeat monitoring.

The second type of research is based on an audio monitoring. In every cardiac cycle, two distinctive sounds (i.e., first and second heart sounds) are generated. Kranjec [8] established a sound acquisition experiment based on a condenser microphone in quiet surroundings and confirmed its validity. However, similar to the RADAR monitoring method, extracting the heart sound from high noise requires further study in actual driving.

The third type of research shows that heartbeat signals can be measured by unconstrained methods by piezoelectric and capacity sensors, including installing sensors on the seat belt [11], steering wheel, and driving seat [12]. Lim [9] attached an acceleration sensor to a chair, and the experiment exhibited excellent performance. Recent studies have shown that the heartbeat signal can be effectively extracted from the pressure sensor. Chen [13] demonstrated a flexible hollow microstructure-enhanced pressure sensor, which can be used to detect the heartbeat signal under the body weight in a noncontact mode. Tohara [14] applied the impulse response signal of a pressure sensor on one heartbeat to estimate the sleep state. Heartbeat monitoring using pressure sensors with a high resolution can even be used to detect the fetal heart rate by locating pressure sensors on a belt worn by the mother [15]. Additionally, a sitting posture can be recognized by sensors array [16], thus the frequency of body shifting can be used to evaluate physiological fatigue [17]. Therefore, both psychological and physiological fatigue can be monitored by pressure sensors, indicating that this is a promising method for practical applications. As the optimal monitoring point is changed following the inevitable movement of the driver’s body, the correlation between electrocardio signals and sensor signals for heartbeat extraction needs to be clarified due to the diversity of the sitting posture. The optimum searching of the monitoring point requires further research into applying the heartbeat-based fatigue monitoring method in a practical context.

To solve the above-mentioned problem, an experiment was conducted to attach a flexible pressure sensors array to the backrest of the car seat. As the sensors array is fabricated with 256 monitoring points to form a 16 × 16 matrix, the optimal heartbeat measure point can be chosen by driving posture classification without moving the pressure sensors array. Recently, there has been growing interest in the development of classification methods for driving posture recognition, such as k-nearest neighbours (KNN) [5], multi-layer perceptron (MLP) [18], convolutional neural network (CNN) [19], fuzzy support vector machine (SVM) [20], etc. Among these methods, extreme learning machine (ELM) [21] has been proven to have an excellent performance in terms of classification accuracy. ELM has two evident advantages that outperform SVM. ELM analytically obtains the weight of the output layer instead of the iterative training process similar to SVM, which makes the learning extremely faster than SVM. Inspired by the fact that SVM with the kernel method achieves good success, ELM could also apply the kernel method. It was proven that ELM is not only more efficient, but also slightly better in terms of accuracy than SVM [22].

In this paper, based on the extreme learning machine (ELM) classification method, six typical driving postures were recognized by the distribution of pressure signals. Then, the heartbeat signal extracting method was analyzed. A band-pass filter in the heart rate range was adapted to all of the pressure signals in the frequency domain. Furthermore, the peak point array of the processed pressure spectrum was derived and had the same distribution as the pressure signals. This conclusion proved that heartbeat signals can be extracted from pressure sensors array. Then, the correlation coefficient analysis of processed pressure signals and electrocardio-signals was performed. The optimal monitoring point was selected from 256 sensors with the highest level of correlation. Finally, the effects of posture on the correlation coefficient were discussed, and the optimal heartbeat measure point was chosen by the unconstrained monitoring method.

It should be noted that heartbeat signals are easily polluted by body movements and vehicle vibration. Noise reduction be a subject of further research, and the signal-to-noise ratio of the heartbeat signal will greatly improve and make the pressure signal based heartbeat extraction more practical.

## 2. Experiment

### 2.1. Requirements for the Subjects

Ten healthy male subjects were involved in the test with controlled conditions. The ages of the subjects ranged from 20 years old to 25 years old. Their heights ranged from 169.08 cm to 176.16 cm, and their weights ranged from 59.21 kg to 74.03 kg. The experiment began at 9:00 am to ensures the normal heartbeat signals were obtained in the most energetic time of the day. All the subjects were asked to rest well during the previous night. Tobacco, wine, tea, coffee, and any other food and drugs that might affect the heart rate were forbidden during the previous day.

### 2.2. Experiment Equipment and Setting

#### 2.2.1. Equipment for Obtaining Electrocardio-Signals

A PC-80B high-speed electrocardio-signal detector (Figure 1) was used to collect electrocardio-signals during the experiment. The electrodes were pasted as illustrated Figure 2.

#### 2.2.2. Equipment for Obtaining Pressure Signals

A SR SVZA4545L pressure sensors array manufactured by Tokai rubber industries, LTD in Komaki, Japan (Figure 3a) was used to collect pressure signals for the heartbeat signal extracting. In Figure 3b, the pressure sensors array was set between the driver and backrest of the car seat. The 16 × 16 pressure sensors array is arranged by 256 sensors, and the available measuring area is 250 mm × 250 mm. The sensors were numbered from 1 to 256, as shown in Figure 3c. The frequency of the electrocardio-signals was concentrated from 0.5 Hz to 2 Hz [6]. Accordingly, the sampling frequency for the pressure signal was set to 30 Hz.

The simulated driving test was performed on the SCANeR simulation training system manufactured by OKTAL, LTD in France (Figure 4), which provides a virtual 3D urban road traffic scene for the driving simulation experiment. The electrocardio-signal detector and pressure sensors array were installed as demonstrated in Figure 5.

### 2.3. Experimental Program and Data Acquisition

1. Driving posture pattern recognition experiment based on the distribution of pressure signals: Six types of typical driving postures were selected and the pressure distributions were recorded. Then, the pattern recognition algorithm of ELM was used to recognize the driving posture.

2. Pressure sensors array monitoring for heartbeat signal extraction: The signals for the pressure sensors array and electrocardio-signals for experimental verification were determined. The driving environment was simulated by the SCANeR training system. Electrocardio-signals and pressure sensors array signals were measured in the above mentioned six types of sitting postures. Then, heartbeat signals were extracted based on the pressure signals. The correlation between the pressure based heartbeat signals and electrocardio-signals was analyzed in six driving postures.

## 3. Driving Posture Calcification by Extreme Learning Machine

### 3.1. Selection of Typical Driving Posture

A postural angle model is shown in Figure 6. Cervical flexion, elbow angle, hip angle, and knee angle were selected as dependent variables to evaluate the driving posture [20].

According to the subjective questionnaire, the first four comfort ranges of the postural angle were chosen as driving postures 1–4 and are summarized in Table 1 based on the questionnaire.

In order to discuss the applicability of the unconstrained monitoring method for heartbeat signals measurement under an unsuitable sitting posture, the body left leaning at 10° and the body right leaning at 10° in the driving posture 1 (postural angle A1:145–155,A2: 115–120,A3: 102–106,A4: 124–130) were chosen as driving postures 5 and 6. Take subject 1 as an example, the driving posture and pressure distribution in driving posture 1 is shown in Figure 7. The color maps of the pressure distribution were measured by the 16 × 16 pressure sensors array.

### 3.2. Brief Review of Extreme Learning Machine

There are *N* training samples, (*x_i_*, *t_i_*)*_i_*_=1,…,*N*_. *x_i_* ∈ *R^n^* consists of the input data of *n* dimensions, and *t_i_* ∈ {1, …, *m*} is a label data of *m* dimensions. The hidden nodes of ELM are set to $\tilde{N}$ and the output function is *G*(*a_i_*, *b_i_*, *x*). So, the ELM model can be described as:(1)$$f_{\tilde{N}}\left(x_{j}\right)=\overset{\tilde{N}}{\underset{i=1}{\sum }}\beta_{i}G\left(a_{i},b_{i},x_{j}\right)=t_{j},j=1,\ldots ,N$$
where the input weight, *A* = [*a*_1_, *a*_2_, …, $a_{\tilde{N}}$], and bias, *B* = [*b*_1_, *b*_2_, …, $b_{\tilde{N}}$], are parameters of the hidden layer, and *β_i_* is the output weight of the *i*th hidden node used to connect with the output node. Equally, (1) can be written as:(2)Hβ=T
where:(3)$$H\left(a_{1},\ldots ,a_{N},b_{1},\ldots ,b_{\tilde{N}},x_{1},\ldots ,x_{N}\right)={\left[\begin{array}{ccc} G\left(a_{1},b_{1},x_{1}\right) & \cdots & G\left(a_{\tilde{N}},b_{\tilde{N}},x_{1}\right)\\ \vdots & \ddots & \vdots \\ G\left(a_{1},b_{1},x_{N}\right) & \cdots & G\left(a_{\tilde{N}},b_{\tilde{N}},x_{N}\right) \end{array}\right]}_{N\times \tilde{N}}$$
(4)$$\beta ={\left[\begin{array}{c} \beta_{1}^{T}\\ \vdots \\ \beta_{\tilde{N}}^{T} \end{array}\right]}_{\tilde{N}\times m},\;\mathrm{and}\;T={\left[\begin{array}{c} t_{1}^{T}\\ \vdots \\ t_{N}^{T} \end{array}\right]}_{N\times m}$$

*H* is the output matrix of the hidden layer of ELM, and the *i*th column of *H* is the output of the *i*th hidden node.

In ELM, *H* can be obtained according to the training set and the randomly assigned parameters (*a_i_*, *b_i_*) of the hidden layer. Then, the output weight, *β*, of ELM is calculated as:(5)$$\hat{\beta }=H^{*}\times T$$
where $H^{*}$ is the Moore-Penrose generalized inverse of the matrix, *H*. The ELM algorithm is shown as follows.

Take training set, {(*x_i_*,*t_i_*)|*x_i_* ∈ *R^n^*,*t_i_* ∈ *R^m^*, *i* = 1,…,*N*}, the hidden layer output function, *G*(*a_i_*,*b_i_*,*x*), and the number of hidden nodes, $\tilde{N}$ as inputs. The ELM algorithm can be divided into the following steps:Assign input weight, *A* = [*a*_1_, *a*_2_, …, $a_{\tilde{N}}$], and bias, *B* = [*b*_1_, *b*_2_, …, $b_{\tilde{N}}$], randomly.Calculate the hidden layer output matrix, *H*.Calculate the output weight, *β*: $\hat{\beta }=H^{*}\times T$ [23].

### 3.3. Feature Extraction of the Pressure Distribution Image

#### 3.3.1. Image Processing before Feature Extraction

The image processing is described as follows:(1)Pressure distribution images were treated by gray level transformation;(2)Noise reductions were carried out by the median filtering method;(3)Gray images were converted to a binary one by the suitable threshold value. Additionally, the edges were detected by the binary image.(4)The minimum enclosing rectangles were extracted.

#### 3.3.2. Feature Parameters Selection and Calculation

Nineteen feature parameters in three categories were extracted from the pressure distribution image, including (1) the geometric feature calculated by the minimum enclosing rectangle, including the circumference, area, roundness, and the invariant moment, HU (1)–HU (7); (2) the texture feature calculated by the gray level co-occurrence matrix, including the mean of energy, standard deviation of energy, mean of entropy, standard deviation of entropy, mean of inertia moment, and standard deviation of inertia moment; and (3) the color characteristics, including the mean value of the red, green, and blue color [24].

A normalized process is performed by using Formula 6 after obtaining the 19 feature parameters:*Q_i_*^*^ = *Q_i_/max|Q_i_|*(6)

*Q_i_* represent the 19 feature parameters mentioned above, and *Q_i_^*^* is the normalized eigenvalues.

### 3.4. Accuracy Evaluation of ELM by Training Samples Size and Hidden Node Selection

To evaluate the accuracy of ELM by selecting the training samples size and hidden node, we tested the six postures for a continuous two hours of monitoring. In this study, 10 subjects were monitored and 20 groups of data were selected for each of the six driving postures, that is, a total of 1200 sets of data were acquired. Take sigmoid as the activation function, the numbers of training samples was randomly selected from 100 to 800, and 100 were added at a time. Meanwhile, the number of hidden nodes was selected from 30 to 120 and increased in every 10.

In Figure 8, with the increase of the sample size, the accuracy rate increases rapidly when the sample size is from 100 to 500. Then, the rising rate tends to slow down from 600 to 800. A considerable accuracy increase is also shown when the hidden nodes are in the range of 30 to 70. In the range of 80 to 120, the accuracy shows a flat change. Therefore, it seems more reasonable to choose a sample size of 600 and a hidden node size of 70, because this selection takes into account both the accuracy and computational speed. The accuracy reaches 95% under these circumstances, which satisfies the accuracy requirements for classification.

The pressure distributions of driving postures 1 to 4, as shown in Figure 7, have minor differences, thus extracting more eigenvalues can improve the classification accuracy. In addition, the literature [16] points out that ELM do not consider the weight of each sample in the training set, which may cause the accuracy to decrease, especially in imbalanced datasets. The improved algorithms need to be discussed in further study.

## 4. Extraction of Heartbeat Signals from the Pressure Sensor Matrix

In the time domain, 1 min of 16 × 16 pressure matrix signals were acquired. Figure 9 illustrates the pressure signal distribution of the driving posture 1 through a mean value. The maximum signals in Figure 9 are caused by a close contact to the driver’s seat.

A band-pass filter in the heart rate range was adapted to the pressure signals in the frequency domain. For example, the heart rate of subject 1 during the test was from 75 bpm to 80 bpm; that is, the corresponding frequency was 0.75–0.8 Hz. The pressure data from number 1–256 were band filtered into the frequency domain in the range of 0.75–0.8 Hz. Furthermore, the power spectrum peak values of the 256 processed signals were derived. In Figure 9, nine measuring points were selected and their power spectrum peak values are plotted at the corresponding locations. Five of them have pressure values greater than 2 V, one is between 1 V and 2 V, and three are less than 1 V. It is obvious that the power spectrum peak value in the frequency domain has the same distribution as the pressure signals in the time domain. These phenomena confirm that the heartbeat signals can be extracted from the pressure signals.

IFFT (inverse fast Fourier transform) was used to obtain the processed time domain signal. Figure 10 depicts the time domain signals of the maximum pressure point in number 181. Figure 11 exhibits the time domain signals in number 181 after processing. The electrocardio-signals are monitored simultaneously, as displayed in Figure 12. The heartbeat signal cycle is the same as the electrocardio-signals presented in Figure 12 despite the evident noise that is demonstrated in Figure 11.

## 5. Correlation Analysis of Heartbeat Signals and Electrocardio-Signals

The correlation of the two sets of data between electrocardio-signals and heartbeat signals was analyzed.
(7)$$\left|\rho xy\right|=\frac{\sum_{i=1}^{n}\left(x_{i}-\bar{x})(y_{i}-\bar{y}\right)}{\sqrt{\sum_{i=1}^{n}{\left(x_{i}-\bar{x}\right)}^{2}\cdot \sum_{i=1}^{n}{\left(y_{i}-\bar{y}\right)}^{2}}}$$
where, |ρxy| is the correlation coefficient, *x_i_* and *y_i_* are the two sets of data to evaluate the degree of concordance. $\bar{x}$ and $\bar{y}$ are the mean value of *x_i_* and *y_i_*, respectively.

The correlation coefficient, |ρxy|, of the electrocardio-signals with 256 sets of heartbeat signals in driving posture 1 was calculated and is displayed in Figure 13. In accordance with the definition of the correlation coefficient, |ρxy|=1 indicates a strong correlation between two signals, and |ρxy|= 0 when the two signals are independent. In Figure 13, the correlation coefficient between electrocardio-signals and 256 measurement points on the pressure sensor are varied. The correlation coefficient in measuring points 0–50 and 200–256, where the driver’s back is not in close contact with the backrest, is less than 0.4, thereby exhibiting a non-correlation. However, the correlation function values in measuring point 50–200, where the driver’s back is in close contact with the backrest, are in the range of 0.8–1. The highest correlation coefficient was 0.99 in the maximum pressure signal point 181 (Figure 9) given the intimate contact between the driver and backrest where the pressure sensors were set up. Therefore, the optimum monitoring points can be obtained by searching the monitoring points with the maximum correlation coefficient.

## 6. Accuracy Analysis of Heartbeat Signal Extraction in Varied Sitting Postures

To obtain the optimal measure point considering the varied driving postures, the correlation coefficient between the electrocardio-signals and heartbeat signals are illustrated in Figure 14. From driving postures 1–4 in the y-coordinate, the change in the maximum pressure point caused by the driver’s posture corresponds to the maximum correlation coefficient. Thus, the heartbeat signals in the four driver postures can be extracted from the pressure signal. In driving postured 5 and 6, the correlation coefficients in all measuring points were below 0.4, thereby indicating that electrocardio-signals can be hardly extracted in the two driving positions. Table 2 presents the measuring point optimization selection in varied driving postures in accordance with the correlation coefficient analysis.

## 7. Conclusions

In this study, an unconstrained method for monitoring the driver’s heartbeat signal based on the pressure sensor was investigated. The following conclusions were obtained through the driving posture calcification, heartbeat signal extraction, and correlation analysis:

(1) The method for heartbeat signal extraction from the pressure sensors array was analyzed. IFFT was used to transform the time domain pressure signals to the frequency domain. Then, these signals were band filtered in the range of the heart rate. The mean pressure signals had the same distribution as the power spectrum values of the processed pressure signals. This result confirms that the heartbeat signals can be extracted from the pressure matrix signals. The vibration cycles of the electrocardio-signals and band-filtered pressure signals after the IFFT transformation were similar.

(2) The analysis of the correlation coefficient acquired in driving posture 1 indicates that the correlation coefficient in 50–200 measure points were in the range of 0.8–1. The high correlation coefficient was caused by the close contact of the driver’s back and pressure sensors array attached to the backrest. In particular, the correlation coefficient in point 181 was up to 0.99, which shows a high correlation between the pressure signal and electrocardio-signals. The heartbeat signals extracted from the maximum correlation measured points can be used to monitor driving fatigue instead of electrocardio-signals.

(3) The correlation coefficient between the electrocardio-signals and heartbeat signals in varied sitting postures was analyzed. Studies have indicated that the heartbeat signals can be extracted in driving postures 1–4. The other two driving positions were unsuitable for extracting heartbeat signals. The optimum measuring point in varied sitting postures was selected as the conclusion drawn from this study.

(4) Obtaining a heartbeat signal with a high signal-to-noise ratio is the first step to monitoring driving fatigue. This research provides an unconstrained monitoring method, which can select optimal monitoring points in different driving postures effectively. In future research, we will carry out noise reduction research caused by vehicle vibration in actual driving, and put the unconstrained driving fatigue monitoring to practical use.

## Figures and Tables

**Figure 1 sensors-19-00368-f001:**
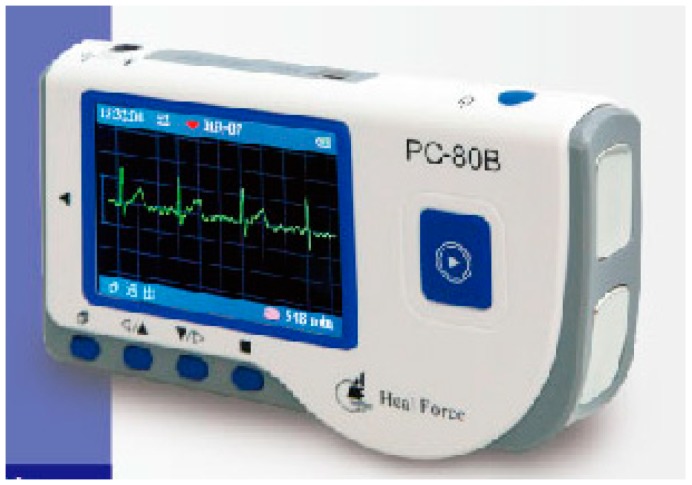
PC-80B high-speed electrocardio-signal detector.

**Figure 2 sensors-19-00368-f002:**
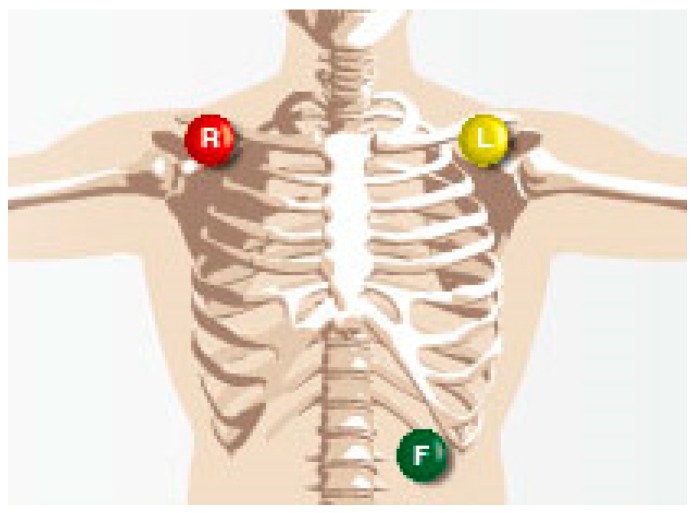
Location of the electrode paste.

**Figure 3 sensors-19-00368-f003:**
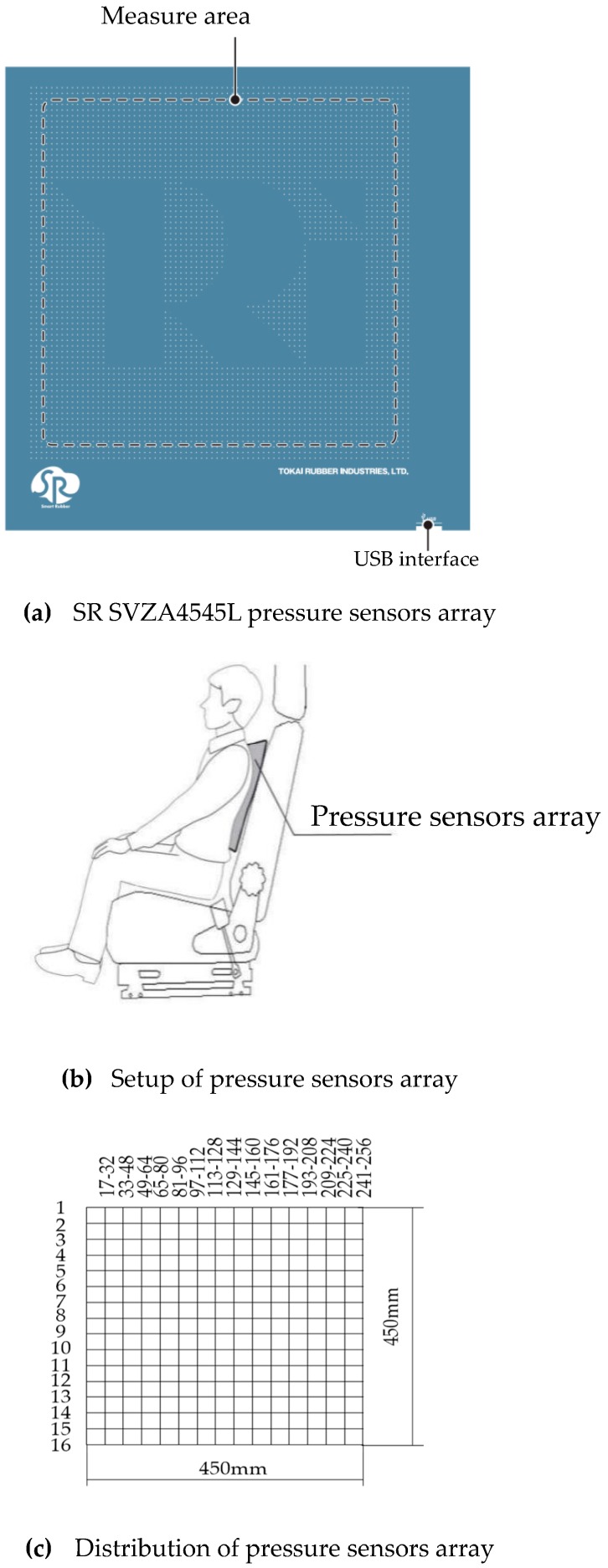
Schematic of pressure sensors setup.

**Figure 4 sensors-19-00368-f004:**
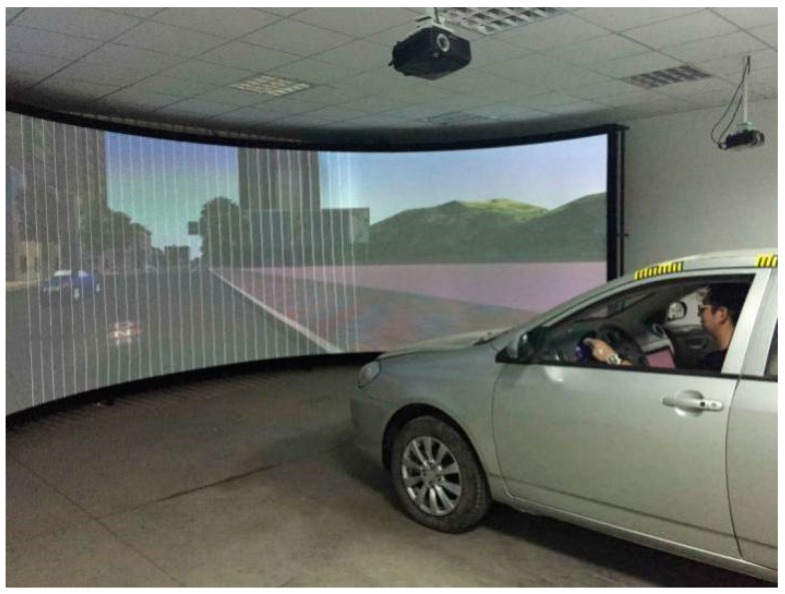
SCANeR simulation training system.

**Figure 5 sensors-19-00368-f005:**
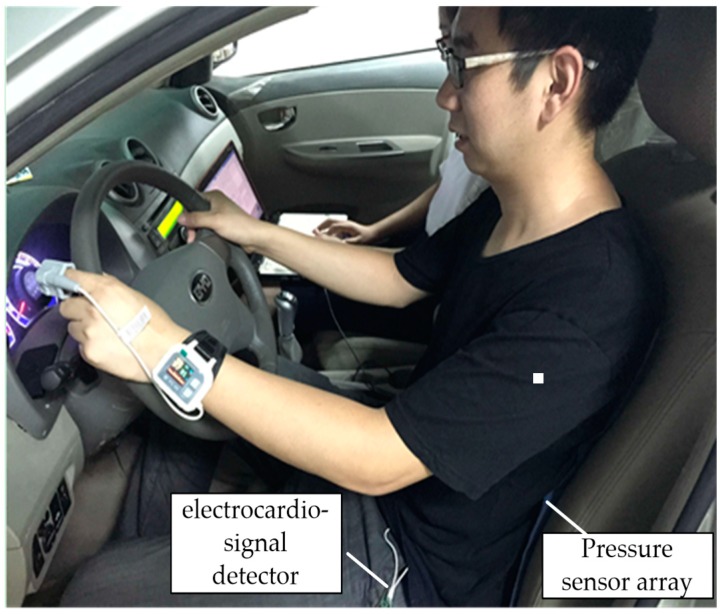
Equipment installation.

**Figure 6 sensors-19-00368-f006:**
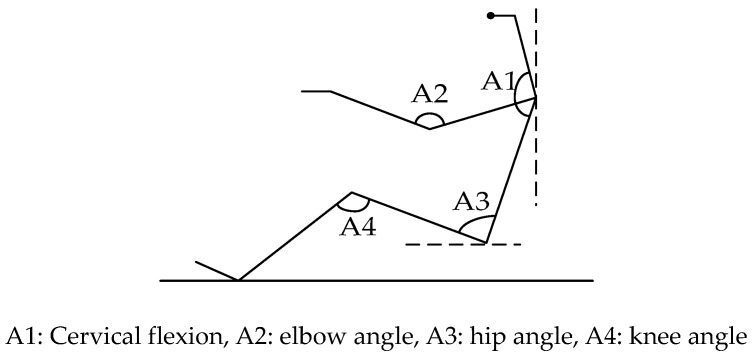
Postural angle model.

**Figure 7 sensors-19-00368-f007:**
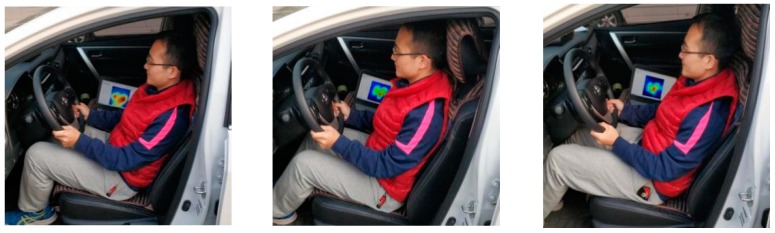
Sitting posture and pressure distribution of subject 1.

**Figure 8 sensors-19-00368-f008:**
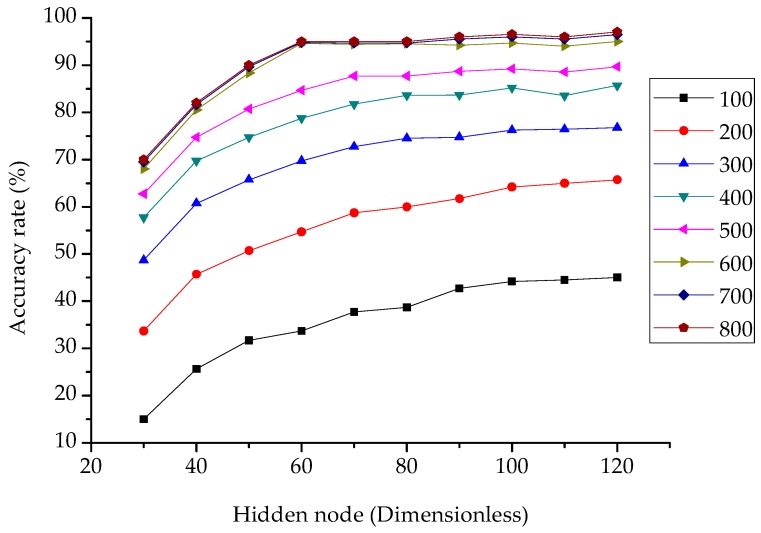
The results of ELM with different sample sizes and hidden nodes.

**Figure 9 sensors-19-00368-f009:**
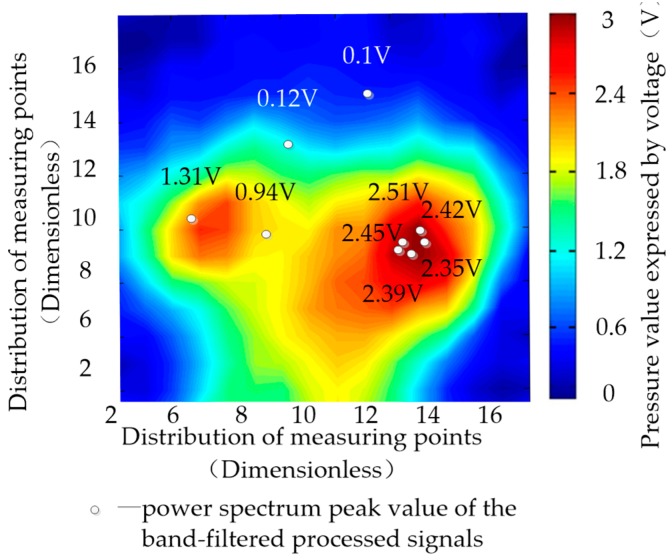
Pressure contour and electrocardio-signal extraction.

**Figure 10 sensors-19-00368-f010:**
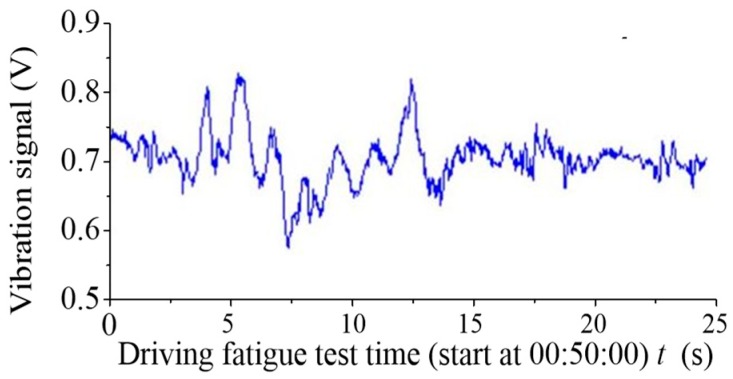
Time domain signal of number 181 in the vibration sensor array.

**Figure 11 sensors-19-00368-f011:**
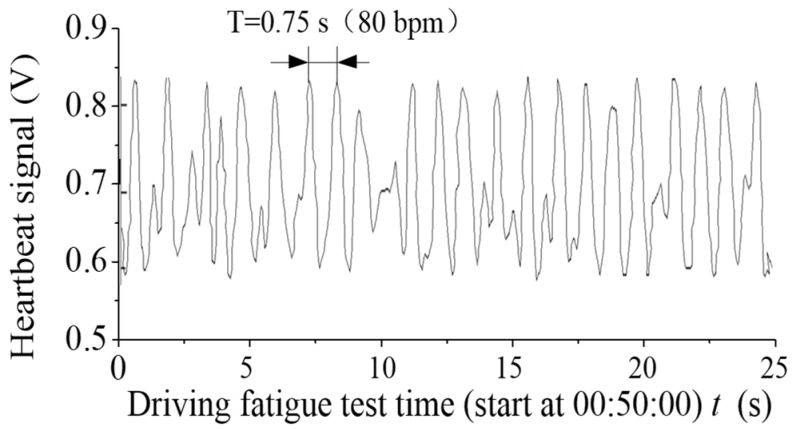
Extracted heartbeat signals of number 181 by noise reduction.

**Figure 12 sensors-19-00368-f012:**
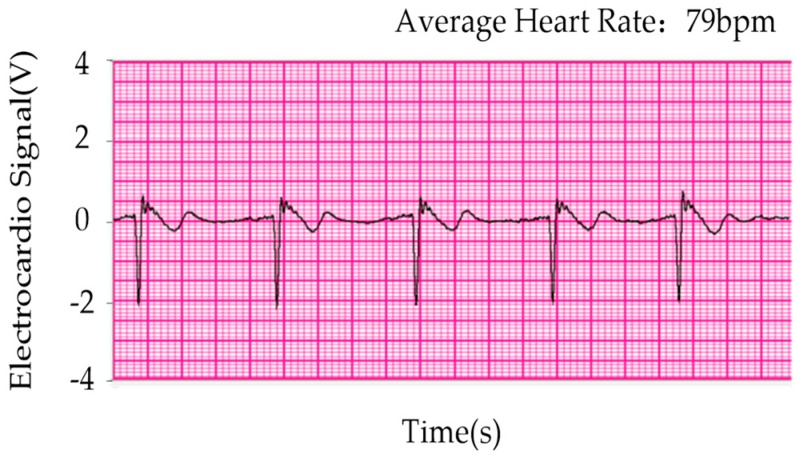
Electrocardio-signal recording.

**Figure 13 sensors-19-00368-f013:**
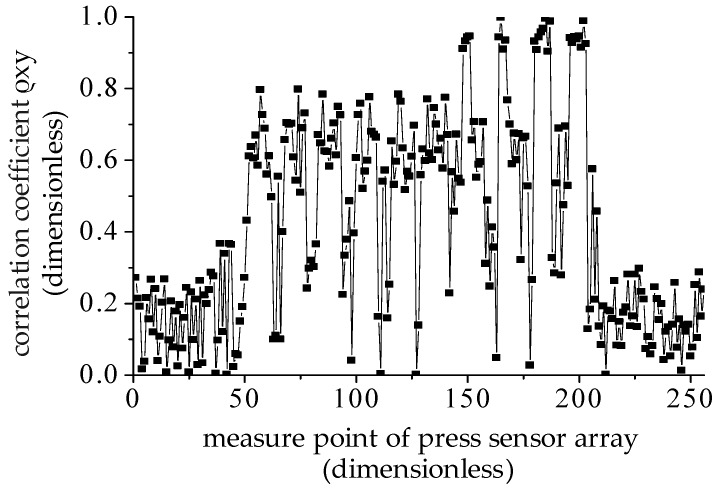
Correlation coefficient analysis of heartbeat signals and electrocardio-signals.

**Figure 14 sensors-19-00368-f014:**
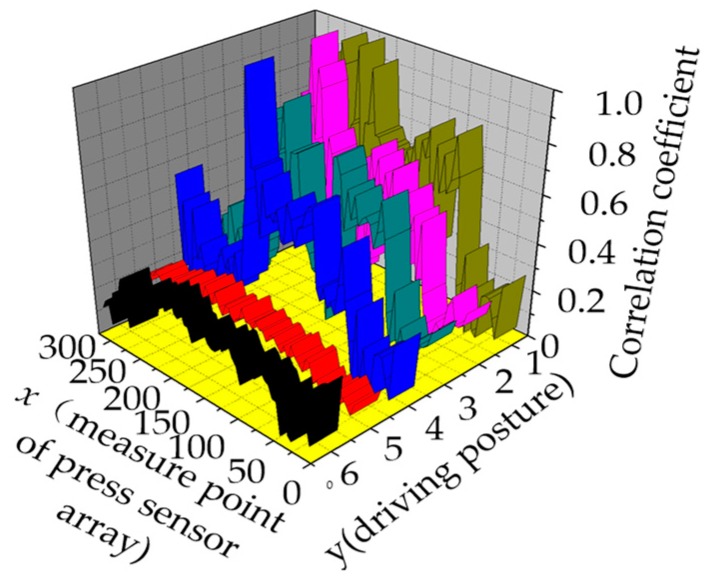
Correlation coefficient analysis in varied driving postures.

**Table 1 sensors-19-00368-t001:** Range of comfort and typical driving posture for the postural angle.

Postural Angle	Range of Comfort (°)	Driving Posture 1 (°)	Driving Posture 2 (°)	Driving Posture 3 (°)	Driving Posture 4 (°)
A1: Cervical Flexion	130–160	145–155	130–135	135–145	155–160
A2: Elbow Angle	92–153	115–120	114–120	116–122	112–116
A3: Hip Angle	99–115	102–106	98–102	102–108	109–110
A4: Knee Angle	112–139	124–130	124–130	124–130	124–130

**Table 2 sensors-19-00368-t002:** Measuring point optimization selection.

Sitting Position	Heartbeat Signals Can Be Extracted	Monitoring Point	Correlation Coefficient
driving posture 1	√	181	0.99
driving posture 2	√	186	0.92
driving posture 3	√	202	0.94
driving posture 4	√	185	0.89
driving posture 5	×	×	×
driving posture 6	×	×	×

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
