# Peer review of "Unconstrained Monitoring Method for Heartbeat Signals Measurement using Pressure Sensors Array"

_sensors, 2019, doi:10.3390/s19020368_

Round 1

Reviewer 1 Report

Major Comments:

It is important to justify the choice of support vector machine (SVM) classification method. There are many other classification methods. It is recommended to review other classification methods such as Extreme Learning Machine (ELM), k-nearest neighbours (KNN), multi-layer perceptron (MLP), neural networks (NN) and other classification methods.  These methods should be well reviewed and discussed.

More details of the implementation support vector machine (SVM) classification method and its results should be provided. 

Limitations of the method should be appropriately discussed and addressed.

There should be a discussion on how the results/ accuracy can be further improved.

Author Response

We would like to thank you for the thorough review of our manuscript "Unconstrained monitoring method for heartbeat signals measurement using pressure sensors array" by Yongxiang JIANG, Sanpeng DENG, Sunhong CHANG, QiYu MING.

As requested we have provided a comprehensive description of the changes that we have made to the manuscript to address the reviewers' comments (pages following this cover). We believe that we have been able to fully address all the reviewers' concern. In addition, we herein attach the revised manuscript. All the changes of the manuscript are printed in RED color.

The type of our paper for peer review was set up by editor before sending, the Figure number 4~6 went wrong and Figure 12 was replace by a wrong Figure.We revised it back.

Thank you very much for your consideration and we look forward to your final decision.

1.It is important to justify the choice of support vector machine (SVM) classification method. There are many other classification methods. It is recommended to review other classification methods such as Extreme Learning Machine (ELM), k-nearest neighbours (KNN), multi-layer perceptron (MLP), neural networks (NN) and other classification methods.  These methods should be well reviewed and discussed.

Response:

Following the reviewer's suggestion, we chose Extreme Learning Machine classification methods for driving posture recognition.(see line 146 to line 223 of the revised manuscript)

2.More details of the implementation support vector machine (SVM) classification method and its results should be provided.

Response:

Following the reviewer's suggestion, more details of the implementation Extreme Learning Machine classification methods is discussed.(see line 166 to line 223 of the revised manuscript)

3.Limitations of the method should be appropriately discussed and addressed.

Response:

Following the reviewer's suggestion, several classification methods are discussed , Limitations of the SVM and the advantage of ELM are discussed.(see line 70 to line 83 of the revised manuscript)

4.There should be a discussion on how the results/ accuracy can be further improved.

Response:

Following the reviewer's suggestion, we discussed the results improved method for further study. (see line 219 to line 223 of the revised manuscript)

Reviewer 2 Report

The abstract is an important section of a paper. Its main goal is to present in a clear and concise way, the central idea of the work and its novelty. The authors must improve the abstract section in this sense. They must focus their explanations.

It would be interesting that the authors motivate that the results obtained and presented in this paper are limited to application of specific characteristics explained.

In the fig. 12 it would be necessary to use the english language.

It would be necessary to introduce more references concerning with methods used for facing

this problem.

Author Response

We would like to thank you for the thorough review of our manuscript "Unconstrained monitoring method for heartbeat signals measurement using pressure sensors array" by Yongxiang JIANG, Sanpeng DENG, Sunhong CHANG, QiYu MING.

As requested we have provided a comprehensive description of the changes that we have made to the manuscript to address the reviewers' comments (pages following this cover). We believe that we have been able to fully address all the reviewers' concern. In addition, we herein attach the revised manuscript. All the changes of the manuscript are printed in RED color.

The type of our paper for peer review was set up by editor before sending, the Figure number 4~6 went wrong and Figure 12 was replace by a wrong Figure.We revised it back.

Thank you very much for your consideration and we look forward to your final decision.

1. The abstract is an important section of a paper. Its main goal is to present in a clear and concise way, the central idea of the work and its novelty. The authors must improve the abstract section in this sense. They must focus their explanations.

Response:

Following the reviewer's suggestion, we revised the abstract and in particularly focused on description the novelty of this paper and the specific methods. (see line 14 to line 26 of the revised manuscript)

2. It would be interesting that the authors motivate that the results obtained and presented in this paper are limited to application of specific characteristics explained.

Response:

Following the reviewer's suggestion, we discussed the limited to application of ELM classification methods and the results improved method for further study .(see line 219 to line 223 of the revised manuscript)

3. In the fig. 12 it would be necessary to use the english language.

Response:

Our paper for peer review was simple edited by Editor before sending, we inquired the reason to the editor and she said something went wrong when she made simple layout in the paper for peer review, Figure 12 in the paper for peer review is shown as follows

We revised it back to the original Figure 12 .(see line 268~269 of the revised manuscript).

Figure 13. Correlation coefficient analysis of heartbeat signals and electrocardio-signals

4.It would be necessary to introduce more references concerning with methods used for facing this problem.

Response:

Following the reviewer's suggestion, we introduced more references concerning with the methods of heartbeat extracting from pressure sensor. The problem of optimum seeking of monitoring point is discussed.(see line 47~52,61~66,70~83 of the revised manuscript)

The reference added:

8.Zhang, D.; Wang,H.;Wu,D.Toward Centimeter-Scale Human Activity Sensing with Wi-Fi Signals. Computer.2017, 50(1),48-57,doi: 10.1109/MC.2017.7.

9.Tomimori, H.; Sano, S.; Nakano, Y. Development of adaptive noise reduction technology for in-vehicle heartbeat sensor.Proceedings of the 2011 7th International Conference on Intelligent Sensors, Sensor Networks and Information Processing.2011, 25-29,doi:10.1109/ISSNIP.2011.6146570.

10.Tenenbaum, C.;Haynes, D.;Pham, P.;et al. Integrated Sensor Technologies Preventing Accidents Due to Driver Fatigue. Faculty.uml.edu.1970.

11.Tohara, T.;Katayama, M.;Takajyo, A.; et al. Time frequency analysis of biological signals during sleep.SICE Annual Conference 2007, 2007, 1925-1929, doi: 10.1109/SICE.2007.4421301

12.Zhao, C.; Gao, Y.; He, J.; et al. Recognition of driving postures by multiwavelet transform and multilayer perceptron classifier. Engineering Applications of Artificial Intelligence.2012, 25(8),1677-1686, doi:10.1016/j.engappai. 2012.09.018

13.Yan, C.;Zhang, B.; Coenen, F. Driving posture recognition by convolutional neural networks.International Conference on Natural Computation.2016, 10(2),103–114 doi:10.1049/iet-cvi.2015.0175

14.Wu, Q.; Luo, S.; Sun. S.;A Computer-Aided Driving Posture Prediction System Based on Driver Comfort.International Conference on Advances in Artificial Reality & Tele-existence.2006,1088 – 1097, doi:10.1007/11941354_113

15.Huang,G.B.;Zhu,Q.Y.;Siew,C.K.Extreme learning machine: Theory and applications. Neurocomputing.2006, 70(1-3):489-501,doi:10.1016/j.neucom. 2005.12.126.

16.Wang,L.L.; Cheng,Y.;Li, W.; Chen,J.B. Hyperspectral Image Classification by AdaBoost Weighted Composite Kernel Extreme Learning Machines.Neurocomputing.2018, LNCS275,1725–1733, doi:10.1016/ j.neucom. 2017.09.004

In addition,REVIEWER #3 suggest us to delete the irrelevant information about (1) physical movements and (2) driving actions. We also add a new unconstrained monitoring method using Wi-Fi RF signals ,all the changes of the introduction are printed in RED color .

Reviewer 3 Report

In this paper, the author used a pressure sensor array to monitor driving fatigue. The paper did not provide sufficient information or sound arguments, thus not suitable to be published in Sensors.

The author introduces a lot of information that is not relevant to their research. I suggest the author directory introduce the advantage and disadvantage of his method, which belongs to category 3, i.e., physiological information method. 

There is not enough information about the pressure sensor array. Did the author fabricate their own sensor array? Or where did the author buy such sensor array? 

Ten subjects are not enough to do classification.

The designed six posture cannot reflect the real scenario during driving. 

In Figure 12, please use English axes. 

Author Response

We would like to thank you for the thorough review of our manuscript "Unconstrained monitoring method for heartbeat signals measurement using pressure sensors array" by Yongxiang JIANG, Sanpeng DENG, Sunhong CHANG, QiYu MING.

As requested we have provided a comprehensive description of the changes that we have made to the manuscript to address the reviewers' comments (pages following this cover). We believe that we have been able to fully address all the reviewers' concern. In addition, we herein attach the revised manuscript. All the changes of the manuscript are printed in RED color.

The type of our paper for peer review was set up by editor before sending, the Figure number 4~6 went wrong and Figure 12 was replace by a wrong Figure.We revised it back.

Thank you very much for your consideration and we look forward to your final decision.

1. In this paper, the author used a pressure sensor array to monitor driving fatigue. The paper did not provide sufficient information or sound arguments, thus not suitable to be published in Sensors.

Response:

We have thoroughly thought over the specific opinions put forward by the reviewers and have made our best to revise the paper. In the introduction section, we add researches about heartbeat extraction from pressure sensors(see line 61~62 of the revised manuscript). We also emphasize that the purpose of this paper is focus on unconstrained monitoring method for heartbeat signals measurement using pressure sensors array(see line 64~66 of the revised manuscript), driving fatigue monitoring is our further study. We hope to get your guidance and the opportunity to publish the paper.

2. The author introduces a lot of information that is not relevant to their research. I suggest the author directory introduce the advantage and disadvantage of his method, which belongs to category 3, i.e., physiological information method.

Response:

Following the reviewer's suggestion,we delete the irrelevant information about (1) physical movements and (2) driving actions . In addition, we introduced a new unconstrained monitoring method using Wi-Fi RF signals(see line 47~52 of the revised manuscript).More references concerning with the methods of heartbeat extracting from pressure sensor is discussed(see line 61~66 of the revised manuscript). The disadvantage of unconstrained heartbeat extraction methods is discussed (see line 63~66). To solve the above mentioned problem, the optimum seeking of monitoring point is pointed out, the methods for driving posture recognition is introduced. (see line 70~83 of the revised manuscript)

3. There is not enough information about the pressure sensor array. Did the author fabricate their own sensor array? Or where did the author buy such sensor array?

Response:

Following the reviewer's suggestion, the information about the pressure sensor array is described. (see Figure 3, line 113~114  of the revised manuscript)

4.Ten subjects are not enough to do classification.

Response:

Reviewer 1 also proposed a revision to the support vector machine method. He thought we need to justify the choice of support vector machine (SVM) classification method, the limitations of SVM and the accuracy could not meet the need for driving posture classification in only ten subjects. Reviewer 2 point out that posture pattern recognition should not conducted on a different seat. Following the reviewer's suggestion, we have justify the classification method,new experiments have been done. We tested the six postures for continuous two hours monitoring. In this study, 10 subjects were monitored and 20 groups of data were selected for each of six driving posture, that is, a total of 1200 sets of data were acquired. 800 samples (150 samples in each of 6 driving postures) were randomly selected as training samples for ELM. The remaining 400 samples were testing samples. (see line 202~209 of the revised manuscript)

5.The designed six posture cannot reflect the real scenario during driving.

Response:

Following the reviewer's suggestion, the reasons for choosing six driving postures are elaborated in detail.(see line 147~163 of the revised manuscript)

6.In Figure 12, please use English axes.

Response:

Our paper for peer review was simple edited by Editor before sending, we inquired the reason to the editor and she said something went wrong when she made simple layout in the paper for peer review,Figure 12 in the paper for peer review is shown as follows

We revised it back to the original Figure 12 .(see line 91~98 of the revised manuscript).

Figure 13. Correlation coefficient analysis of heartbeat signals and electrocardio-signals

Reviewer 4 Report

1. In text citations in introduction go up to 26 with some numbers missing, but reference list goes only up to 13.

2. Line 132: Please clarify the meaning of "The test was conducted from 9:00 am to 11:00 am." Did subjects drive for 2 hours? Later the paper says the test was for 30 seconds and 1 min.

3. Figure 3a: Sensor array is shown behind the backrest, not between the driver and backrest. The position of the sensor array contradicts figure 4 (or 5?). One figure shows the array near the upper back and the other near the waist. Exact distance from the bottom of the backrest was not given in the paper.

4. Figure 3b: Linear index is used for 2D array of sensors, but after row 1 the order of number allocations to the sensors is not clear. The size of the sensor array or distances between the sensors were not given in the paper.

5. Text refers to the wrong figure number for figures 5, 6. There is no figure 6 in the paper and there are two figure 4.

6. More information on the model/brand and type of the sensors used, distance between them and location relative to the car seat is required to be able to reproduce the experiment. Was the full model information of the SCANeR simulator given? 

7. Posture pattern recognition was conducted on a different seat. How can authors be sure that the results are valid for the actual simulator seat? The seat heights, shapes, tilts and materials are different.

8. Please be more specific about what "standard sitting position" and other positions mean. Both body and backrest can have different tilt.

9. Results in figure 5 have no dimensions or intensity scales, it is not mentioned if the colour scales are the same or not in terms of the intensities that they represent. What parameter was measured exactly, the range of measurement values and units of measurements are not given. Similar issue with figure 8.

10. Please provide the reference for feature extraction and origin of formulas 1-3.Table 1 has 9 values per group, please clarify how they were obtained from formula 1. Formula 3 does not have C in it, please clarify and add a description for K, x as well.

11. The sentences "Comparing the pressure signal distribution with the power spectrum peak value, the maximum power spectrum peak value is in the place where the maximum time domain signals are located. Moreover, the distribution of the minimum and intermediate values is the same" are confusing, please clarify. Similar issue with "The distribution of the mean pressure signal and power spectrum values of the band-filtered-processed signals is the same" in line 286.

12. Line 227 and figure 9:  Is maximum pressure point number 18 or 181?

13. Text in figure 11 is too small.

14. Please add formula for correlation coefficient in relation to measured parameters.

15. Figure 12 has non English text on axes, meaning of LF was not explained, the time scale goes up to 80 min while it was previously mentioned that the measurements were taken for 30 seconds, the y axis does not correspond to expected values of 0-1 for correlation coefficient. Did the authors insert the correct figure?

16. Table 2: point numbers disagree with the figure 13, where the highest correlation is around ~200.

17. The claim that study can be used to monitor fatigue is more of a future work suggestion. Tests were not conducted till the driver reaches fatigue and there are no comparisons of driver with fatigue and without. The possibility of heart rate monitoring was demonstrated.

Author Response

We would like to thank you for the thorough review of our manuscript "Unconstrained monitoring method for heartbeat signals measurement using pressure sensors array" by Yongxiang JIANG, Sanpeng DENG, Sunhong CHANG, QiYu MING.

As requested we have provided a comprehensive description of the changes that we have made to the manuscript to address the reviewers' comments (pages following this cover). We believe that we have been able to fully address all the reviewers' concern. In addition, we herein attach the revised manuscript. All the changes of the manuscript are printed in RED color.

The type of our paper for peer review was set up by editor before sending, the Figure number 4~6 went wrong and Figure 12 was replace by a wrong Figure.We revised it back.

Thank you very much for your consideration and we look forward to your final decision.

1. In text citations in introduction go up to 26 with some numbers missing, but reference list goes only up to 13.

Response:

We are sorry to make such a "terrible" mistake. Modification has been made for text citations in introduction. As the other three reviewers put forward a large number of modification requirements in the introduction part, this part This part has changed a lot, and we made careful modifications and checked the references one by one.

2. Line 132: Please clarify the meaning of "The test was conducted from 9:00 am to 11:00 am." Did subjects drive for 2 hours? Later the paper says the test was for 30 seconds and 1 min.

Response:

The description of the experimental time is intended to emphasize 9:00 am to 11:00 am is the most energetic time of a day and the heartbeat signals without abnormal state can easily obtained from subjects. In the experiment on patterns recognition of driving posture, we tested the six postures for continuous two hours monitoring., 10 subjects were monitored and 20 groups of data were selected for each of six driving posture, that is, a total of 1200 sets of data were acquired. The above explanation have been added into the paper. (see line 99~100,202~204 of the revised manuscript).

3. Figure 3a: Sensor array is shown behind the backrest, not between the driver and backrest. The position of the sensor array contradicts Figure 4 (or 5?). One Figure shows the array near the upper back and the other near the waist. Exact distance from the bottom of the backrest was not given in the paper.

Response:

We setup the sensor array between the driver and backrest.it can be shown in both Figure 5 and Figure 7 of the revised manuscript ,We apologize for the confusion caused by the wrong position of pressure sensor array in Figure 3(Schematic of pressure sensor setup).Modification has been made in Figure 3, we presents the position of the pressure sensor array between the driver and backrest.

The purpose of the experiment is to extract the heartbeat signal regardless the different driving posture. As the size of the pressure sensor matrix is large enough, it can be ensure that at least one point of the pressure sensor in the 16*16 sensors array is nearby the heart ,thus, the novelty of this paper is to adjust the monitoring point according to the change of driving posture without moving the pressure sensors array.the distance from the bottom of the backrest is not limited in the experiment. The use of pressure sensor array is to solve this problem and ensure the universal applicability in this method. Inspired by reviewers,in the abstract and introduction, we emphasize the purpose and significance of the research, we pointing out that the purpose of the research is to adjust the optimal monitoring point from time to time due to the differentdriving posture of the driver. The above explanation have been added into the paper. (see line 15~16 and line 70~73 of the revised manuscript)

4.Figure 3b: Linear index is used for 2D array of sensors, but after row 1 the order of number allocations to the sensors is not clear. The size of the sensor array or distances between the sensors were not given in the paper.

Response:

Following the reviewer's suggestion, The sensor of each line is numbered. The size of the sensor array were also given in Figure 3c. (see Figure 3c of the revised manuscript)

5. Text refers to the wrong Figure number for Figures 5, 6. There is no Figure 6 in the paper and there are two Figure 4.

Response:

Our paper for peer review was simple edited by Editor before sending, we inquired the reason to the editor and she said something went wrong when she made simple layout,we revised Figure number 4 to 6 back to the correct number.

6. More information on the model/brand and type of the sensors used, distance between them and location relative to the car seat is required to be able to reproduce the experiment. Was the full model information of the SCANeR simulator given?

Response:

Following the reviewer's suggestion,The information of pressure sensor array and SCANeR simulator have been given. (see line 113~114,128~130 and reference of the revised manuscript).

7. Posture pattern recognition was conducted on a different seat. How can authors be sure that the results are valid for the actual simulator seat? The seat heights, shapes, tilts and materials are different.

Response:

Following the reviewer's suggestion, new experiments have been done on the driving seat, we revised Figure 7. (see Figure 7 of the revised manuscript).

8.Please be more specific about what "standard sitting position" and other positions mean. Both body and backrest can have different tilt.

Response:

Following the reviewer's suggestion, we added more specific about six positions.(see line 148~163 of the revised manuscript)

9.Results in Figure 5 have no dimensions or intensity scales, it is not mentioned if the colour scales are the same or not in terms of the intensities that they represent. What parameter was measured exactly, the range of measurement values and units of measurements are not given. Similar issue with Figure 8.

Response:

Following the reviewer's suggestion,we revised Figure 9 of the revised manuscript. The color maps of pressure distribution in Figure7 is described according to the reviewer's suggestion. (see line 162 ,Figure7 and Figure 9 of the revised manuscript).

10. Please provide the reference for feature extraction and origin of formulas 1-3.Table 1 has 9 values per group, please clarify how they were obtained from formula 1. Formula 3 does not have C in it, please clarify and add a description for K, x as well.

Response:

Reviewer #1 suggest us to change support vector machine (SVM) classification method to other classification methods. We revised the third section to a new classification method more suitable for driver's sitting posture pattern recognition. (see line 147~222 of the revised manuscript).

11. The sentences "Comparing the pressure signal distribution with the power spectrum peak value, the maximum power spectrum peak value is in the place where the maximum time domain signals are located. Moreover, the distribution of the minimum and intermediate values is the same" are confusing, please clarify. Similar issue with "The distribution of the mean pressure signal and power spectrum values of the band-filtered-processed signals is the same" in line 286.

Response:

Following the reviewer's suggestion, we revised the two sentences.(see line 227~237,291~293 of the revised manuscript).

12. Line 227 and Figure 9:  Is maximum pressure point number 18 or 181?

Response:

The maximum pressure point number is 181.We are sorry to make such a "terrible" mistake due to carelessness. We examined the whole paper carefully and modification has been made.(see line239,246,248, 265,of the revised manuscript and table 2).

13. Text in Figure 11 is too small.

Response:

Following the reviewer's suggestion, we revised Figure 11.(see Figure 12 of the revised manuscript).

14. Please add formula for correlation coefficient in relation to measured parameters.

Response:

Following the reviewer's suggestion, we add formula for correlation coefficient.(see line 253~255 of the revised manuscript).

15. Figure 12 has non English text on axes, meaning of LF was not explained, the time scale goes up to 80 min while it was previously mentioned that the measurements were taken for 30 seconds, the y axis does not correspond to expected values of 0-1 for correlation coefficient. Did the authors insert the correct Figure?

Response:

Our paper for peer review was simple edited by Editor before sending, we inquired the reason to the editor and she said something went wrong when she made simple layout in the paper for peer review, Figure 12 in the paper for peer review is shown as follows

We revised it back to the original Figure 12 .(see line 91~98 of the revised manuscript).

Figure 13. Correlation coefficient analysis of heartbeat signals and electrocardio-signals

16. Table 2: point numbers disagree with the Figure 13, where the highest correlation is around ~200.

Response:

We carefully checked all the mistake of point numbers in highest correlation coefficient. Modification has been made in Table 2.

17. The claim that study can be used to monitor fatigue is more of a future work suggestion. Tests were not conducted till the driver reaches fatigue and there are no comparisons of driver with fatigue and without. The possibility of heart rate monitoring was demonstrated.

Response:

Heartbeat signal is an effective method for fatigue monitoring as stated in introduction. Therefore, the purpose of this study is focused on an unconstrained method for extracting heartbeat signal and to adjust heartbeat measurement place according to driving posture. Following the reviewer's suggestion, we add prospects for future research.(see line 307~311 of the revised manuscript).

Round 2

Reviewer 1 Report

I congratulate the authors for some significant improvement in the manuscript. However, it needs to be improved further better addressing the concerns raised. I would recommend to provide better justification justifying the choicce of sample size and how it would ensure the significance of the results.

Author Response

Following the reviewer's suggestion, we evaluate the accuracy of ELM by training samples size and hidden node selection.We also explained the choice of sample size takes into account both accuracy and computational speed.(see line 224 to line 238 of the revised manuscript)

Reviewer 2 Report

Minor revision (corrections to minor methodological errors and text editing)

Author Response

Thank you for reviewing our paper and affirming our revision. I examined the paper carefully and revised the methodological errors and text editing in the article. (see line 4,line 14~26, 41, 68, 81, 82, 207~211, 306 of the revised manuscript)

Reviewer 3 Report

I appreciate the author's response. Due to my limited knowledge, I still don't think the pressure sensor array is a good way to monitoring driving fatigue. The pressure sensor is a deformation induced sensitive device. A lot of motion artifacts during driving will affect the result.

Author Response

Following the reviewer's suggestion,we add more introduction, focus on the purpose and advantage of choosing pressure sensor. Several reference were added to confirm the possibility for using pressure sensors array for monitoring driving fatigue. We also mentioned the further research direction ,that is noise reduction of heartbeat signal.(see line  33 ~ 40, 70 ~ 80, 109~112 of the revised manuscript)

Driving fatigue includes three categories:

(1) Periodic fatigue caused by insufficient sleep and overtime work;

(2) Physiological fatigue caused by static force work of muscle;

(3) Psychological fatigue caused by high intense of nerve centre.[1]

Accordingly, currently developed monitoring methods can be divided as follows:

(1) Visual-based fatigue monitoring, such as blinking, nodding, facial expression changes, head movement, eye motion;

(2) Driving actions, such as behaviors in steering, braking, acceleration and , body “slumping,” or body shifting; 

(3) Physiological information, including electroencephalo-, electromyo-, and electrocardio-signals and respiration.[2]

Recent studies have shown that the heartbeat signal can be effectively extracted from the pressure sensor,Chen[3] demonstrated a flexible hollow microstructure-enhanced pressure sensor, which can be used to detect heartbeat signal under the body weight in a noncontact mode. Tohara[4] applied impulse response signal of pressure sensor on one heartbeat to estimate sleep state. Heartbeat monitoring using pressure sensors with a high resolution can even be used to detect the fetal heart rate by locating pressure sensors on a belt worn by mother[5].Additionally, sitting posture can be recognized by sensors array[6] , thus, the frequency of body shifting can be used to evaluate physiological fatigue.Therefore, both psychological and physiological fatigue can be monitored by pressure sensors, and make this a promising way for practical application. 

It should be noted that, heartbeat signal are easily polluted by body movements and vehicle vibration . Noise reduction will carry out for further research, signal-to-noise ratio of heartbeat signal can greatly improve and make the pressure signal based heartbeat extraction into practical.

Some references were added, please check the attachment.
